# Identification of Inflammatory and Regulatory Cytokines IL-1α-, IL-4-, IL-6-, IL-12-, IL-13-, IL-17A-, TNF-α-, and IFN-γ-Producing Cells in the Milk of Dairy Cows with Subclinical and Clinical Mastitis

**DOI:** 10.3390/pathogens11030372

**Published:** 2022-03-17

**Authors:** Zane Vitenberga-Verza, Māra Pilmane, Ksenija Šerstņova, Ivars Melderis, Łukasz Gontar, Maksymilian Kochański, Andżelika Drutowska, Gergely Maróti, Beatriz Prieto-Simón

**Affiliations:** 1The Institute of Anatomy and Anthropology, Rīga Stradiņš University, 1010 Rīga, Latvia; mara.pilmane@rsu.lv (M.P.); ksenija.serstnova@rsu.lv (K.Š.); ivars.melderis@bkus.lv (I.M.); 2Research and Innovation Centre Pro-Akademia, 95-050 Konstantynów Łódzki, Poland; lukasz.gontar@proakademia.eu (Ł.G.); maksymilian.kochanski@proakademia.eu (M.K.); andzelika.drutowska@proakademia.eu (A.D.); 3Seqomics Biotechnology Ltd., 6782 Morahalom, Hungary; marotig@seqomics.hu; 4Biological Research Center, Plant Biology Institute, 6726 Szeged, Hungary; 5Department of Electronic Engineering, Universitat Rovira i Virgili, 43007 Tarragona, Spain; beatriz.prieto-simon@urv.cat; 6ICREA, 08010 Barcelona, Spain

**Keywords:** mastitis, bovine, cytokines, inflammation, immunohistochemistry, milk, diagnosis

## Abstract

In naturally occurring bovine mastitis, effects of infection depend on the host inflammatory response, including the effects of secreted cytokines. Knowledge about the inflammatory and regulatory cytokines in milk cells of free-stall barn dairy cows and in naturally occurring mastitis is lacking as most studies focus on induced mastitis. Hereby, the aim of the study was to determine inflammatory and regulatory cytokines in the milk of dairy cows with subclinical and clinical mastitis. The following examinations of milk samples were performed: differential counting of somatic cells (SCC), bacteriological examination, and immunocytochemical analysis. Mean SCC increased in subclinical and clinical mastitis cases. The number of pathogenic mastitis-causing bacteria on plates increased in subclinical mastitis cases but decreased in clinical mastitis. The inflammatory and regulatory markers in the milk cells of healthy cows showed the highest mean cell numbers (%). In mastitis cases, immunoreactivity was more pronounced for IL-4, IL-6, IL-12, IL-13, IL-17A, TNF-α, and IFN-γ. Data about subclinical and clinical mastitis demonstrate inflammatory responses to intramammary infection driven by IL-1α, IL-4, and IL-17A. Moreover, the host defense response in mastitis is characterized by continuation or resolution of initial inflammation. IL-12 and INF-γ immunoreactivity was recognized to differ mastitis cases from the relative health status.

## 1. Introduction

Bovine mastitis is extremely important due to its economic, social, and environmental impact. This infectious disease is not limited to animal health and welfare but can also question potential endangerment of dairy consumers. The economic burden of bovine mastitis on the dairy sector is huge due to production losses and processing [1]. In addition, mastitis of dairy cows contributes significantly to increasing antimicrobial resistance at the forefront of the most pressing global health challenges suggesting misdiagnosis and reduction in antibiotic usage [2,3,4,5]. So far, delayed and poor intervention of bovine mastitis is mainly due to significant limitations in the methods currently used for diagnosis, including their expenses [6]. On-farm diagnosis of mastitis mainly relies on somatic cell count, although this method depends on multiple factors such as the type of causative pathogen and inner factors [7]. The study builds upon new knowledge acquired on host response-derived biomarkers, such as proinflammatory cytokines, identified as prospective mastitis biomarkers to report disease status and etiology. Leveraging the presence of these biomarkers in milk, a "liquid biopsy" approach could be pursued to ensure non-invasive, stress-free, and prompt diagnosis of naturally occurring bovine mastitis.

Mastitis is mainly a bacterial infection; however, it is also caused by algae, viruses, fungi, or improper milking procedures [8,9,10,11,12,13]. *Prototheca* spp. is the third most common mastitis pathogen after *Streptococcus* and *Staphylococcus* spp. [14]. Mastitis can manifest into clinical or subclinical conditions. Recent evidence suggests that effects of infection depend on the host response in the early stages of the disease, including the effects of secreted cytokines [15,16,17]. Cytokines are small molecule proteins that play an important role in cell-to-cell communication. There is a myriad of processes that are stimulated or inhibited by cytokines, such as cell differentiation, proliferation, remodeling, degeneration, regeneration, and even cell death. Both experimentally induced or naturally occurring mastitis results in an increase in the somatic cell count (SCC) and levels of produced cytokines (interleukin (IL) IL-1, IL-2, IL-4, IL-5, IL-6, IL-8, IL-10, and IL-12) in milk [18]. Similarly, also biochemical analysis of milk provides plenty of diagnostic options in search of mastitis markers [19].

The interleukin (IL)-1 (IL-1) family is one of the most widely represented groups of cytokines [20]. IL-1 family cytokines work in relation to the tissue immunocompetent cells via several signaling pathways [21], performing mostly inflammatory, autoimmune, fibrosis-initiating, and mitogenic functions [22]. In experimental acute bovine mastitis, interleukin-1 alpha (IL-1α) demonstrates local modulation of such first-line defense mediators such as prostaglandin (PG-F2α) and leukotriene (LT-B_4_), suggesting its role in early inflammation [23].

The main producers of IL-4 in bovine mammary glands are T and B lymphocytes, eosinophils and basophils, mast cells, plasma cells [24,25], as well as epithelial cells, which together form the basis of type 2 immunity [26]. Furthermore, IL-4 regulates innate immunity and houses an inhibitory effect on interferon (IFN)-γ in dairy cows [25]. However, associated with mastitis, both IL-4 expression [27] and levels in milk [28] were decreased, suggesting further studies.

IL-6 is a pleiotropic cytokine that has plenty of different functions such as the host organism defense by performing immune response, hematopoiesis, and regulation of inflammation [29,30]. In bovine mastitis, IL-6 is considered as an early but nonspecific indicator of various inflammation states, especially in subclinical mastitis [28]. Contrary, IL-6 was not significantly different between healthy and subclinical mastitis-affected cows questioning its association with other factors [31].

IL-12 stimulates IFN-γ production and NK cell proliferation, cytotoxicity, and induces NK cells to produce cytokines. In addition, IL-12 promotes the proliferation and activation of T cells and provides immune cell differentiation [32]. IL-12 has been studied to increase in the milk of cows with naturally occurring mastitis, suggesting its role in innate immunity [33].

Mast cells and basophils as key effector cells of innate immunity participate in allergic proinflammatory responses by secreting soluble factors of type I cytokine IL-13 [34]. IL-13 in bovine udder, milk, and association with mastitis has been poorly studied.

IL-17A is a proinflammatory cytokine that has an important role in host defense against different microbial and non-microbial pathogens. IL-17A is produced by multiple cell types of both the adaptive and innate immune systems [35,36,37,38]. In induced bovine mastitis, IL-17A mediates host defense-pathogen interactions during mastitis [39,40].

Tumor necrosis factor (TNF)-α is a proinflammatory cytokine mainly produced by macrophages. Depending on the localization of its release and the receptor it will bind to, it can perform different functions, such as stimulating the synthesis of other cytokines and causing inflammatory reactions, controlling vital processes of the cell, and maintaining tissue homeostasis [29,41,42,43,44]. IFN-γ is a cytokine secreted by different cells of the innate and adaptive immunity, as well as antigen-presenting cells having the features of protective immune responses and elimination of pathogens [45,46,47,48]. Altogether with other selected cytokines, TNF-α and IFN-γ multifunctionality in bovine subclinical and clinical mastitis must be studied.

In the bovine mammary gland, mRNA and relative protein expression levels indicate the up-regulation of proinflammatory cytokines (e.g., TNF-α and IL-6) in infected tissues rather than non-infected tissues. Ignited proinflammatory pathways implicate the underlying regulatory pathways for proper immune function in mammary glands [49]. Various molecules released by the host in the early stages of infection have already been identified as biomarkers for early diagnosis. Interestingly, their expression has previously been reported to be related to the causative agent [17]. Altogether, studies on the cytokines in bovine milk promote a better understanding of the role of these mediators in the pathogenesis of mastitis. This knowledge suggests potential therapeutic applications and the identification of diagnostic and prognostic biomarkers.

Hereby, the aim of the study was to determine inflammatory and regulatory cytokines IL-1α, IL-4, IL-6, IL-12, IL-13, IL-17A, TNF-α, and IFN-γ in the milk of dairy cows with subclinical and clinical mastitis, and to differentiate the changes of their expression throughout three consecutive days after the settlement of the diagnosis of subclinical or clinical mastitis. Knowledge about the inflammatory and regulatory cytokines in milk cells of free-stall housed dairy cows and in naturally occurring mastitis is lacking as most studies focus on pathogen-induced mastitis. In naturally occurring mastitis, the study of cytokines in milk is a potential tool for the early and timely diagnosis and in prospective pathogenesis-based treatment.

## 2. Results

### 2.1. Evaluation of Milk Quality

In healthy cows, the mean SCC log10 values were 4.65 (0.13) cells on day 1, 4.44 (0.26) cells on day 2, and 4.45 (0.14) cells on day 3 (Figure 1). In subclinical mastitis-affected cows, mean log10 (SD) SCC numbers were 5.37 (0.34) cells on day 1, then increased to 5.90 (0.34) cells on day 2, but then decreased to 5.77 (0.52) cells on day 3. In the clinical mastitis group, SCC mean log10 SCC values were increasingly rising from 6.04 (0.37) cells on day 1 to 6.35 (0.33) cells on day 2 and to 6.43 (0.10) cells on day 3.

Statistical analysis determined that mean log10 value of SCC did not statistically differ over day 1 to 3 in healthy animals (*p =* 0.076), in subclinical mastitis-affected cows (*p =* 0.074), and in clinical mastitis-affected cows (*p =* 0.07). The main effect of status indicated a statistically significant difference in mean log10 SCC counts between study groups (F(2, 12) = 78.308, *p* < 0.001, partial η^2^ = 0.929). Mixed two-way ANOVA determined a statistically significant two-way interaction of status and time on SCC mean log10 value (F(4, 24) = 3.491, *p =* 0.022, partial η^2^ = 0.368). Mean log10 SCC values were statistically significantly greater in clinical mastitis group than in healthy cows (*p* < 0.001) and in subclinical mastitis-affected mastitis cows (*p =* 0.004), and also statistically significantly greater in subclinical mastitis group than in healthy cows (*p* < 0.001).

### 2.2. Bacteriological Examination

Microbiological examination evaluated the logarithmic values of the total number of pathogenic mastitis-causing bacteria (log CFU/mL) or the absence of bacterial growth (bacterial clearance).

In subclinical mastitis-affected cows, mean TBC log CFU/mL increased from a value of 2.80 (0.47) on day 4 to a value of 3.47 (0.46) on day 5, to a value of 3.53 (0.43) on day 6 (Figure 2). In milk of clinical mastitis-affected cows, mean TBC log CFU/mL values were 3.95 (0.66) on day 4 and 3.95 (0.67) on day 5, but then decreased to a value of 3.59 (0.75) on day 6.

In subclinical mastitis, a statistically significant overall difference was evaluated between days 4 to 6 (F(2, 8) = 4.6, *p =* 0.047), the population effect size partial omega-squared ω^2^ = 0.4; sphericity assumed); however, post-hoc analysis with a Bonferroni adjustment revealed no statistically significant difference between days (*p* > 0.05). Mixed two-way ANOVA determined no statistically significant two-way interaction of status and time on TBC (*p =* 0.101) in the milk of both subclinical and clinical mastitis-affected cows. In clinical mastitis-affected cows, mean log10 values of TBC did not statistically significantly differ over time (*p =* 0.524).

### 2.3. Identification of Cytokines Producing Cells in Milk of Dairy Cows

In healthy animals, IL-1α immunoreactive cell numbers indicated 1/3 of intensively stained (++) cells and 2/3 of weakly stained (+) cells. Mean (%) IL-1α immunoreactive cell counts in healthy cows decreased from 56.0 cells on day 4 to 49.6 cells on day 5 and to 46.2 cells on day 6. Overall, IL-1α-positive cell counts insignificantly decreased, whereas negative cell counts increased. Immunoreactivity for IL-4 in healthy cows remained stable for most cells being positive; in addition, most of the positive cells were stained intensively (++). These data showed a mean (%) of 96.0 cells on day 4, 98.0 cells on day 5, and 97.2 cells on day 6. Further, IL-6 immunoreactivity in milk cells of healthy cows decreased from day 4 to 6 by mean values (%) of positive cell counts 35.8 on day 4, 33.4 on day 5, and 32.6 on day 6; intensively and weakly stained immunoreactive cells were roughly 1:1. The mean IL-12 immunoreactive cell counts (%) in healthy cows also decreased from 57.2 on day 4 to 53.6 on day 5, and to 51.2 on day 6; in addition, intensively and weakly stained IL-12 immunoreactive cells were near to ratio 1:1. Further, mostly weakly stained (+) IL-13-positive cell counts (%) in healthy cows were low on day 4 with a mean value of 24.6, then increased to a mean value of 26.8 on day 5, whereas on day 6 decreased to a mean value of 21.6. The mean values (%) of mostly weakly stained (+) IL-17A immunoreactive cell counts decreased from 98.2 cells on day 4 to 95.2 cells on day 5 but then increased to 97.2 on day 6. Overall weakly stained (+) TNF-α-positive cell counts indicated an increase in mean cell count (%) from 19.2 cells on day 4 to 20.8 on day 5, followed by a decrease on day 6 to 18.4 cells. Finally, the mean numbers (%) of near 1:1 ratio of intensively (++) and weakly (+) stained IFN-γ immunoreactive cell counts increased from 73.8 cells on day 4 to 76.0 cells on day 5, but then decreased to 69.4 cells on day 6.

Altogether, immunoreactive cell counts in healthy cows were stable throughout day 4 to day 6 for IL-4, decreased for IL-1α, IL-6, and IL-12. However, for IL-13, IL-17A, TNF-α, and IFN-γ, the trend showed an increase in mean immunoreactive cell count from day 4 to 5, followed by a decrease on day 6. Of all factors in healthy animals, the highest mean immunoreactive cell counts were for IL-4, IL-17A, and IFN-γ (Figure 3).

In subclinical mastitis-affected cows, the mean cell counts (%) for mostly weakly (+) stained IL-1α immunoreactive cells increased from 25.8 cells on day 4 to 28.4 cells on day 5, but then decreased to 22.2 cells on day 6. The mean values (%) of mostly weakly stained (+) (ratio 3:1 to intensively stained (++)) IL-4 immunoreactive cell counts in subclinical mastitis-affected cows decreased from 86.0 cells on day 4 to 70.8 on day 5 but then increased to 89.2 cells on day 6 with most cells being positive. Similarly, the mean (%) mostly weakly (+) stained immunoreactive cell counts of IL-6 decreased from 16.2 cells on day 4 to 7.0 cells on day 5 but then increased to 10.6 cells on day 6. In subclinical mastitis-affected cows, the mean 1:1 intensively (++) and weakly (+) stained immunoreactive cell counts (%) of IL-12 increased from 25.6 cells on day 4 to 28.4 cells on day 5 to 36.0 cells on day 6. For IL-13, the immunoreactive cells counts were low and showed a minor increase from 1.2 cells on day 4 to 2.0 cells on day 5 to 3.0 cells on day 6. For IL-17A immunoreactivity in milk mostly weakly (+) stained cells of subclinical mastitis cows, the mean positive cell counts (%) increased from 84.2 cells on day 4 to 84.8 on day 5 to 91.6 on day 6, with most cells being positive for IL-17A. In addition, the mean (%) only weakly stained (+) immunoreactive cell counts for TNF-α increased from 1.6 cells on day 4 to 3.0 cells on day 5, to 5.8 cells on day 6, although they were very low. For IFN-γ in subclinical mastitis cows, the mean immunoreactive cell counts (%) increased from 28.4 cells on day 4 to 39.2 cells on day 5, to 55.2 cells on day 6, and the ratio of intensively (++) to weakly (+) stained cells changed from 1:1 to 1:3.

Overall, immunoreactive cell counts in cows affected by subclinical mastitis increased for IL-12, IL-13, IL-17A, TNF-α, IFN-γ, and IL-4. The mean number of IL-6 immunoreactive cells decreased on day 5 but then increased on day 6. Lastly, the mean numbers of IL-1α immunoreactive cell counts increased on day 5 but decreased on day 6 (Figure 4). In subclinical mastitis-affected cows, the highest means of immunoreactive cell counts were observed for IL-4 and IL-17A.

In clinical mastitis-affected cows, the mean (%) IL-1α mostly weakly (+) stained immunoreactive cell counts increased from 20.8 cells on day 4 to a peak value of 43.4 cells on day 5, but then decreased to 26.0 cells on day 6. For IL-4, the mean (%) mostly weakly stained (+) immunoreactive cell counts in the clinical mastitis group increased from 68.6 cells on day 4 to 85.4 cells on day 5 to 88.0 cells on day 6. Similarly, the mean (%) IL-6 mostly weakly (+) stained immunoreactive cell counts in clinical mastitis-affected cows increased from 9.6 cells on day 4 to 16.2 cells on day 5 to 17.2 cells on day 6. In addition, mean values (%) of IL-12 mostly weakly (+) positive cell numbers increased throughout day 4 to day 6 from 6.8 cells on day 4 to 10.4 cells on day 5 to 11.6 cells on day 6. Mean IL-13 immunoreactive cell counts (%) and overall immunoreactivity for IL-13 in clinical mastitis-affected cows were almost none. In clinical mastitis-affected cows, the mean values (%) of predominant IL-17A weakly stained (+) immunoreactive cell counts increased from 79.6 cells on day 4 to 85.4 cells on day 5 to 88.6 cells on day 6. The mean values (%) of TNF-α weakly stained (+) immunoreactive cell counts in clinical mastitis-affected cows decreased from 3.0 cells on day 4 to 2.4 cells on day 5 to 1.4 cells on day 6. Finally, mean IFN-γ mostly weakly stained (+) immunoreactive cell counts (%) increased from 3.2 cells on day 4 to 2.4 cells on day 5 to 11.8 cells on day 6.

In summary, in clinical mastitis-affected cows, immunoreactive cell mean counts in milk increased for IL-4, IL-6, IL-12, and IL-17A and remained nearly absent for IL-13 (Figure 5). For IL-1α, the pattern of positive cell mean value indicated an increase on day 5 followed by a decrease on day 6. For IFN-γ, a decrease on day 5 was followed by an increase on day 6.

The findings of the mean immunoreactive vs. negative cell counts (%) for all the studied factors are shown in Table 1. The mean values (%) (SD) of immunoreactive cell counts for all factors are depicted in Figure 6.

### 2.4. Statistical Analysis

A one-way repeated-measures ANOVA determined that mean numbers (%) of IL-1α (*p =* 0.253), IL-4 (*p =* 0.728), IL-6 (*p =* 0.892), IL-12 (*p =* 0.614), IL-13 (*p =* 0.442), TNF-α (*p =* 0.842), and IFN-γ (*p =* 0.586) immunoreactive cell counts in healthy cows did not show a statistically significant difference over time. However, the mean numbers (%) of IL-17A immunoreactive cells indicated statistically significant differences between the days 4 to 6 (F(2, 8) = 10.0, *p =* 0.007, the population effect size partial omega-squared ω^2^ = 0.62; sphericity assumed). In particular, post-hoc analysis with a Bonferroni adjustment revealed the mean value of IL-17A immunoreactive cell count was significantly higher on day 4 vs. day 5 and 6 (by 2.0 (95% CI, 0.9 to 3.11) cells, *p =* 0.003, partial η^2^ = 0.909).

For cows with subclinical mastitis, a one-way repeated-measures ANOVA determined no statistically significant difference of mean immunoreactive cell count numbers (%) over time for all factors IL-1α (*p =* 0.294), IL-4 (*p =* 0.085), IL-6 (*p =* 0.081), IL-12 (*p =* 0.494), IL-13 (*p =* 0.147), IL-17A (*p =* 0.084), TNF-α (*p =* 0.294), and IFN-γ (*p =* 0.156).

Similarly, in clinical mastitis-affected cows, no statistically significant difference of mean immunoreactive cell count numbers (%) over time was observed for IL-1α (*p =* 0.068, ε = 0.531), IL-4 (*p* = 0.125), IL-6 (*p* = 0.65), IL-12 (*p* = 0.557), IL-17A (*p* = 0.111), TNF-α (*p* = 0.374), and IFN-γ (*p* = 0.255, ε = 0.51). IL-13 immunoreactivity in clinical mastitis was almost undetectable.

Further analysis to determine the main group effect showed no statistically significant difference in mean (%) IL-1α (*p* = 0.071) and IL-6 (*p* = 0.073) immunoreactive counts between study groups. However, the main status effect showed a statistically significant difference in mean immunoreactive cell counts (%) between studied groups for IL-4 (F(2, 12) = 88.044, *p* < 0.001, partial η^2^ = 0.936), IL-12 (F(2, 12) = 8.95, *p* = 0.004, partial η^2^ = 0.599), IL-13 (F(2, 12) = 32.447, *p* < 0.001, partial η^2^ = 0.844), IL-1A (F(2, 12) = 6.025, *p* = 0.015, partial η^2^ = 0.501), TNF-α (F(2, 12) = 27.523, *p* < 0.001, partial η^2^ = 0.821), and IFN-γ (F(2, 12) = 21.78, *p* < 0.001, partial η^2^ = 0.784).

On day 4 and 6, but not on day 5, mean IL-1α immunoreactive cell counts (%) of healthy animals almost doubled the values found in subclinical mastitis and clinical mastitis-affected animals. Mixed two-way ANOVA determined a statistically significant two-way interaction of status and time on IL-1α immunoreactive cell count (F(4, 24) = 3.633, *p* = 0.019, partial η^2^ = 0.377; sphericity assumed). Mean IL-1α immunoreactive cell count (%) was statistically different between groups on day 4 (F(2, 12) = 9.58, *p* = 0.003, partial η^2^ = 0.615), but not on day 5 (*p* = 0.312) and day 6 (*p* = 0.107). The mean IL-1α immunoreactive cell count in healthy cows on day 4 was statistically significantly greater than in subclinical mastitis- (*p* = 0.012) and in clinical mastitis-affected cows (*p* = 0.004), but not on day 5 (*p* = 0.312) and day 6 (*p* = 0.107).

Altogether, most cells in the milk of studied animals were immunoreactive for IL-4; however, a statistically significant difference was observed between status and time on the mean values (F(4, 24) = 3.198, *p* = 0.031, partial η^2^ = 0.348; sphericity assumed). The mean value (%) of IL-4 immunoreactive cell numbers on day 4 was statistically significantly lower in clinical mastitis-affected cows vs. healthy cows (*p* = 0.006). On day 5, a mean value (%) of IL-4 immunoreactive cell count was statistically significantly lower in subclinical mastitis-affected cows than in healthy cows (*p* < 0.001) and clinical mastitis-affected cows (*p* = 0.026).

For mean (%) IL-6 (*p* = 0.896), IL-12 (*p* = 0.852), IL-13 (*p* = 0.909), IL-17A (*p* = 0.063), TNF-α (*p* = 0.759), and IFN-γ (*p* = 0.184) immunoreactive cell counts, there was no statistically significant interactions found between day and status.

The summary of findings to test the significance of the time point and status on immunoreactive cell counts (%) is shown in Table 2.

## 3. Discussion

Milk SCC is a valuable, easy-to-perform tool in the bovine mastitis diagnostic approach. Combined with clinical objectives, SCC numbers determine subclinical and clinical mastitis. Although low numbers of SCCs in milk are associated with relative health, more detailed observation by differential cell counts showed still existing inflammatory reactions suggesting persistent activity by host defense immunity [50]. This might indicate the host defense reactivity due to the continuous exposure to pathogenic microorganisms in the barn conditions. Reported experimental studies confirm prominent SCC early increase in milk after the intended exposure to mastitis-associated antigens [51], while in vivo studied SCC threshold for subclinical mastitis is also recognized [52]. Previous studies of cows in the dry-off period showed a maximum SCC increase of 12 h after experimental inoculation and continuous increase up to 7 days [53]. On-farm diagnosis of mastitis mainly relies on SCC, although this method depends on multiple factors such as the type of causative pathogen. In subclinical mastitis, SCC greatly differs from the microbiological culture and leukocyte differential studies, suggesting SCC alone might not provide as detailed information of mammary quarter status as leukocyte differential analysis or all together [54]. Laboratory-based diagnosis of mastitis via cell culture and/or molecular analysis effectively identifies bacterial pathogens, but its wide use by farmers is prevented due to the time and/or cost involved. Moreover, SCC alone has been overall criticized for not being sensitive enough to work as a screening tool in subclinical mastitis diagnosis, but it works in the context of other parameters [55,56]. Altogether, microbiological analysis, variety of clinical symptoms, and the search for precise diagnostic approaches request the implementation of multiple parameters or even an algorithm. A knowledge acquired on host response-derived biomarkers in milk, such as proinflammatory cytokines, could potentially support the identification of perspective mastitis biomarkers to report disease status and etiology.

The trend in the presence and findings of inflammatory and regulatory markers in the milk cells of healthy cows showed the highest mean cell numbers (%) compared to subclinical and clinical mastitis-affected cows. If we look at the SCC and proinflammatory cytokines as indicators of early inflammation such as TNF-α, they reach a peak in 1 to 12 h followed by a gradual drop [51], suggesting our findings of inflammatory and regulatory markers on day 4 to 6 in the milk of subclinical and clinical mastitis correspond to the resolution of initial inflammation. During bovine mastitis, CD4+ T lymphocytes, activated upon antigen recognition, dominate. They also activate macrophages through the production of cytokines. Depending on the type of cytokines produced, the Th cells are divided into types Th1 or Th2 [57]. Th1 cells are characterized by the production of IFN-γ and IL-2, whereas Th2 cells produced IL-4, IL-5, IL-6, IL-10, and IL-13 [58]. Importantly, IL-10 inhibits IFN-γ and IL-2 via Th1 lymphocytes; IL-4 and IL-5—via Th2 lymphocytes; IL-1, IL-6, IL-8, IL-12, and TNF-α—via macrophages; IFN-γ and TNF-α—via NK cells [59,60]. In mastitis, IL-10 production highly increases, reaching superior relative numbers over other factors [27]. This may suggest a possible IL-10 anti-inflammatory effect on the reduced numbers of milk cells containing inflammatory and regulatory factors and must be the subject of future research.

Initially, IL-1 was associated with leukocytes and the ability to induce fever in animals [61,62]. Subsequent studies have also described IL-1 family cytokines in relation to the functions of immunocompetent tissue cells such as T cells, macrophages, endothelial and epithelial cells. The effect of IL-1 on immunity is indirect; it works through several signaling pathways involving other messenger molecules [21]. In the mammary gland, IL-1 is produced by macrophages, lymphocytes, monocytes, endothelial cells, and fibroblasts. In addition, IL-1 stimulates its own production, as well as that of IL-6, IL-8, IL-12, and TNF-α [25]. In our study, the IL-1α linear pattern through day 4 to 6 in healthy cows was similar to IL-6, IL-12, and like IL-13, TNF-α, and INF-γ by the drop on day 6, however, having no association with other factors in the cases of subclinical and clinical mastitis. This possibly suggests the key regulation by other proinflammatory cytokines in mastitis conditions. Studies evidence bovine mammary epithelial cells challenged with *E. coli*-derived lipopolysaccharides showed an increase in mRNA expression of IL-1α similar to that found in mastitis-affected cows [63]. When challenged with *E. coli*, *S. aureus,* or *S. agalactiae*, goat mammary epithelial cells also showed increased expression of IL-1α [64]. In epithelial cells of bovine mammary glands, IL-1α effects were associated with the inflammatory response to induced mastitis [65]. In both statuses of subclinical and clinical mastitis, IL-1α mean numbers (%) were statistically significantly lower than in healthy cows, suggesting its role in bovine mammary gland immunity and early inflammation. However, a peak in IL-1α immunoreactivity was noted on day 5 of clinical mastitis-affected cows suggesting its inflammatory role in the support and maintenance of inflammation.

In the bovine milk of healthy cows, the expression of regulatory cytokines IL-4, IL-10, and IFN-γ has been previously reported [66]. In our study, almost all cells in milk samples of healthy cows were indeed positive for IL-4, underpinning a high immunoreactivity. Both IL-4 and IL-13 in the association of differentiation of T cells into Th-2 cells may have an immunoregulatory role in the bovine mammary gland [67]. However, mean IL-13 immunoreactive cell numbers (%) were low in healthy animals and almost absent in animals affected by mastitis. Both mean numbers (%) of IL-4 and IL-13 immunoreactive cells did statistically significantly differ by status, but IL-4 also differed both by status and day, promoting the profound stability of IL-4 both in healthy and disease-affected animals. Activation of bovine tissue macrophages by IL-4/IL-13 induces a cell phenotype that is not anti-inflammatory itself but comprises a diverse phenotype that contributes to the tissue repair and could trigger a proinflammatory reaction. Bovine monocyte-derived macrophages suggested IL-4/IL-13 priming in macrophage polarization to wound-healing phenotype [68]. In bovine endothelial cells, both IL-4 and IL-13 protect the endothelial surface against inflammatory mediator-induced procoagulant changes [69]. This might be crucial to hold readiness for both inflammation risk and tissue damage in healthy cows by both IL-4 and IL-13, but only by IL-4 in mastitis-affected cows. The concentration of IL-4 was significantly lower in the milk of cows with *Staphylococcal* subclinical mastitis compared to control animals [28], possibly supporting our findings by causative explanation in subclinical and clinical mastitis.

Contrary to our findings, IL-6 levels in whey of mastitis-affected cows [70] and subclinical mastitis group cows [28] were shown to be higher than in samples from healthy cows. In addition, the detection of IL-6 concentration in milk samples of subclinical mastitis-affected cows predicted inflammation more precisely and earlier than SCC [71]. In our study, mean IL-6-containing cell numbers (%) did not statistically significantly differ by status or both status and day (time). Up to the knowledge, *Klebsiella pneumoniae* promotes the production of the inflammatory cytokine genes IL-6, IL-8, IL-1β, and TNF-α in mastitis already after 6 h [72]. However, the IL-6 gene was among those genes that were not differentially expressed in healthy vs. mastitis-affected cows suggesting they are not suitable markers or indicators of mastitis [27]. In addition, there was no difference in milk IL-6 between healthy and naturally occurring subclinical mastitis cases [31]. The discrepancy in the findings of IL-6 might substantiate its proinflammatory role in the early inflammation phase. Several experimental studies have shown the importance of inflammatory factors in an early inflammatory event. Within a few h after infusion of *E. coli* lipopolysaccharide, high amounts of IL-1, IL-6, and TNF-α are found in milk that further promote IL-1-induced neutrophil migration [73]. Once bovine mammary epithelial cells recognize microbial components, they can initiate an innate immune response by secreting proinflammatory cytokines (IL-1β, TNF-α, IL-6, IL-8) and antimicrobial molecules (cathelicidin, defensins) [74,75].

In the environment of the mammary gland, IL-12 is known as a mediator between innate and adaptive immunity by the regulation of T lymphocytes. It also works as a growth factor for active NK cells and supports their cytotoxicity [25]. In our study, we found IL-12 immunoreactivity to be the highest in healthy cows compared to lower numbers in subclinical mastitis and the lowest in clinical mastitis. Moreover, IL-12 did not change much through days 4 to 6. Within the relative expression patterns of bovine milk cytokines, IL-12 was found to be significantly higher in late lactation compared to its corresponding level in mid-lactation. Possibly, an increase in IL-12 in the late lactation could indicate the shift toward a more pronounced Th1 immunity [76]. Altogether, elevation of IL-12 may indicate its importance in enhancing the innate immunity responses in bovine udder [76], supporting our results in healthy cows. The cytokine transcriptional pattern of *S. aureus*-associated post-infection IL-12 level presented a significant elevation in 24 h followed by a sharp decrease in 32 h [77]. Similar findings were observed in milk cells of dairy cows experimentally challenged with either *E. coli* or *S. aureus* [78]. However, when challenged with *Pseudomonas aeruginosa*, the maximum peak level was observed after 32 h and remained elevated only for 24 h [79]. In response to *Mycoplasma bovis* intramammary infection, milk IL-12 concentrations increased only within 78 h of infection and remained elevated throughout the study [80]. In *Klebsiella pneumoniae*-challenged mastitis, IL-12 levels peaked at 48 h [81]. Furthermore, the differences in milk IL-12 are associated with lower levels in mild vs. moderate/severe clinical mastitis, Gram-positive vs. Gram-negative microorganisms, early group vs. mid and late days in milk production, and spring vs. other seasons [33], indicating a high heterogeneity of IL-12 peak in mastitis. In our study, the numbers suggested significant differences between the study groups by status, considering IL-12 to be an informative diagnostic tool for mastitis.

For IL-17A, most milk cells were found to be immunoreactive in all studied groups by status. IL-17A is a proinflammatory cytokine with host defense properties against microorganisms both in innate and adaptive immunity and a protective role [82]. After cow immunization with ovalbumin, IL-17A was present in the milk of all the high-responder cows right after milk leukocytosis developed [83]. An increase in IL-17A in response to ovalbumin or its combination with *E. coli* was also studied [84]. In immunized cows with *E. coli*, an increase in TNF-α, IL-6, IL-10, and IL-17A was detected. IL-17A was found in the milk of all cows at the onset of inflammation, and concentrations remained elevated after the acute phase [85]. Therefore, as this occurrence is still high and with increasing pattern in subclinical and clinical mastitis cases of our study, IL-17A significance in the context of acute-phase cytokine might propose not only its continuous role in early defense in healthy animals but also inflammation resolution in mastitis. Certainly, the expression of the IL-17A gene is induced in udder tissues of cows experimentally infected with *E. coli*, suggesting IL-17A could play an important role in mediating host-pathogen interactions during mastitis [39]. In dairy cattle challenged with *S. uberis*, the increase in IL-17A levels matched with inversion of the pre-challenge CD4(+)-to-CD8(+) T lymphocyte ratio, that followed by normalization of this ratio altogether with respective IL-17A levels, suggesting that IL-17A may be involved in the resolution of intramammary infection [86].

TNF-α is an acute-phase cytokine produced during the early stages of infection [87,88]. In our study, we found low numbers of TNF-α-containing immunoreactive cells in milk samples of all studied cases. TNF-α could potentially be associated with early initiative activity; however, this role has been questioned. In serum and whey taken from naturally occurring mastitis-affected animals, significantly higher concentrations of IFN-γ and TNF-α were found. As cows recovered, IFN-γ and TNF-α concentrations reduced significantly [89]. Therefore, TNF-α significance may exceed the proinflammatory role. However, in both serum and milk of mid-lactation cows with feed restriction influences, TNF-α concentrations showed only modest increases during endotoxin infusion [90]. When investigating the effect of intramammarily infused lipopolysaccharide on the acute-phase reaction in early and in late lactation, no detectable levels of TNF-α were found in serum, while levels of both increased in milk. Peak concentration was higher in early lactation than in late lactation [91]. In an experiment of bovine mastitis induced by endotoxin or E. coli, no significant differences between sample types and over time for TNF-α concentrations in milk were detected; however, an increase in TNF-α was observed in afferent lymph [92]. TNF-α immunoreactivity was modest and even weak in our cases, suggesting this factor does not exceed its rather baseline expression and stable presence. Moreover, its functionality is not associated with cell quantity. Lastly, in response to *M. bovis* intramammary infection over a period of several days, initial elevations of TNF-α, IL-1β, and IFN-γ were observed between 90 and 102 h post infection [80], suggesting the late onset of these factors depend on causative pathogens. This was also studied in means of endotoxin amount enough to initiate pronounced cell recruitment but insufficient to induce a marked TNF-α secretion [88]. No significant change in gene expression of IL-2, IL-4, IL-6, IL-8, IL-10, IFN-γ, and TNF-α by real-time PCR in milk among animals without mastitis vs. animals with mastitis was observed [27].

Lastly, IFN-γ indicated a prominent difference between study groups, with most cells being immunoreactive for IFN-γ in healthy animals, increasing moderate numbers of IFN-γ immunoreactive cells in subclinical mastitis cases. However, only a minor number of such cells were found in clinical mastitis. It is known that IFN-γ activates the acquired immune response and T lymphocytes and IL-12 production [93]. The last must be recognized in the context of our findings that marked distinctively similar levels of IL-12 in each status group. Moreover, IFN-γ immunoreactivity also indicated diagnostic potential in mastitis vs. healthy cases. However, IFN-γ significance might be addressed to innate host defense signaling as milk IFN-γ immunoreactivity in our clinical mastitis-affected cows was low. Similarly, the mRNA of IL-2 and IFN-γ was not expressed in milk cells isolated from diseased cows before treatment [94]. In bovine mastitis induced by endotoxin or *E. coli*, IFN-γ concentration was too low to be detected by ELISA [92]. Interestingly, IL-4 maintains an antagonistic action to IFN-γ [27]. Therefore IFN-γ immunoreactive mean numbers were lower in subclinical mastitis and almost absent in clinical mastitis, suggesting IL-4 antagonist activity over INF-γ in mastitis conditions. IL-4 immunoreactivity was prominent in our study.

This study aimed to determine inflammatory and regulatory cytokines in the milk of dairy cows and to differentiate the changes in their expression. Notably, the immunoreactivity for all factors was quite stable throughout the time period of three consecutive days after the settlement of the diagnosis and division of study groups of subclinical or clinical mastitis. Understandably, in naturally occurring mastitis, the study of early inflammatory onset is limited to the time spent on the diagnosis to enroll these cases in the study. Hereby, knowledge, at the cellular and molecular level, of the immune response at relative health and during the infection is fundamental to the early and timely diagnosis and in prospective pathogenesis-based treatment.

Nonetheless, this study has limitations to discuss. The study emphasizes the necessity to extend the time period corresponding to the prolonged acute phase up to 7 to 10 days. The evaluation of cytokines in this time period would express the dynamics of the cytokine appearance in milk. Furthermore, additional to the already studied cytokines, including IL-2, IL-8, IL-10, IL-13, nuclear factor kappa B, beta defensins, transforming growth factor beta 1, and cathelicidin LL37, altogether would explain their relationships in detailed interrelations by correlation analysis to substantiate the knowledge on the broad cytokine pleiad participating in both host immune defense and mastitis inflammation.

## 4. Materials and Methods

### 4.1. Study Animals and Ethical Statements

The research was performed in the herd of 79 cows (Holstein Friesian cattle), which included 41 adult cows (of those, 35 were milking cows) and 38 heifers. Based on the veterinary observations as well as screening of somatic cell count in milk performed over three consecutive days, we selected 15 milking cows and allocated them into three study groups: 5 healthy animals, 5 animals with subclinical mastitis, and 5 animals with clinical mastitis (selection criteria are described in Section 4.3). In healthy and subclinical mastitis-affected groups, the first-, the second-, and the third-parity cows were included. In the clinical mastitis group, the second-, fourth-, and fifth-parity cows were included. Mean parity in each group was 1.6, 2, and 3, respectively, while mean days in milk in each group were 181.6, 149.8, and 256, respectively. Due to previous mastitis infections, three animals had a history of prior veterinary interventions: (1) one cow in the subclinical mastitis group had antibiotic treatment that finished one week before the sampling start; (2) one cow in the clinical mastitis group had non-antibiotic treatment applied (anti-inflammatory ointment) that finished three days before the sampling start; and (3) one cow in the clinical mastitis group had antibiotic treatment that finished one month before the sampling start. Since antibiotic and anti-inflammatory interventions were finished several days before the study start, milk samples from animals subject to those interventions were included in the analyses.

All animals were kept in a free-stall barn located in northern Poland, without access to enclosure or pasture. The cows were milked twice a day and fed two times daily, and water was available ad libitum. The feed included silage, corn silage, and grain by-products, which were provided as separate components.

The study aims to perform an analysis of cow’s milk. The research involving animals use has the clear scientific purpose of increasing their welfare, which justifies their engagement in the research project. The research involved techniques administered in a way that did not cause pain, suffering, distress, or damage to the body of participating animals. In this sense and in view of the national regulations of the partner in charge of validation (RIC Pro-Akademia, Poland), the study did not involve experiments on animals as defined in the Law of 15th January 2015 on the protection of animals used for scientific or educational purposes (Journal of Laws of 2015 pos. 266 with further amendments) [95]. Animal studies were carried out humanely according to national and international Animal Care and Use Committee protocols. The study involved obtaining milk samples by standard milking procedure without affecting the cow’s routine.

### 4.2. Sample Collection

The following sample collection was conducted. On days 1 to 3, milk samples were collected for the milk quality examination (somatic cell count screening) from all milking cows (35 animals, of each 4 quarters). Further, on days 4 to 6, milk samples from selected 15 cows were collected for the milk quality analysis, bacteriological examination, and milk sediment preparations. Samples were collected into sterile containers and immediately transported to the laboratory. Before sample collection, udders were cleaned, dried, and disinfected. The foremilk was discarded before taking the sample. Sample analysis was conducted at RIC Pro-Akademia, Poland. No veterinary treatments (neither antibiotic nor anti-inflammatory) were applied to the animals over the sample collection period.

The following analytical procedures of milk samples were performed: differential counting of somatic cells, bacteriological examination, and immunocytochemical analysis.

### 4.3. Differential Counting of Somatic Cells

Somatic cell count (SCC) is an essential indicator of bovine udder inflammation, which highly correlates with the presence of a mammary infection [96]. SCC is a well-established measurement for identifying animals with suspected mammary gland infection and inflammation [97]. SCC investigation was performed by fluorescent automatic cell counter Lactoscan SCC (Milkotronic Ltd., Stara Zagora, Bulgaria). On days 1 to 3, the research covered all milking cows (*n* = 35, of each 4 quarters) to identify relevant cases of subclinical and clinical mastitis. The animals were assigned to the three study groups using the following criteria, based on the average number of somatic cells in milk and observation of udders/mammary glands, in line with the mastitis severity scoring provided by the scientific annals of the Polish Society of Animal Production (Polskie Towarzystwo Zootechniczne, PZH) [98]: (1) healthy group—SCC under 200,000 cells/mL milk, udders/mammary gland not showing redness, swelling, or pain—equivalent to PZH severity score of I or II; (2) subclinical mastitis group—SCC between 200,000 and 500,000 cells/mL milk, no signs of udder/mammary gland inflammation—equivalent to PZH severity score of III or IV; (3) clinical mastitis group—SCC above 500,000 cells/mL, udders/mammary gland showing redness, swelling, or pain—equivalent to PZH severity score higher than IV. No exclusion criteria for each group were applied.

The further study analyses were aimed to investigate selected udder quarters that were affected by mastitis. In eight cows, one-quarter were affected by mastitis. In two cows with clinical mastitis, out of two mastitis-affected quarters, one quarter was selected by the highest SCC for further analyses.

### 4.4. Bacteriological Examination

Microbiological analysis was performed according to ISO 4833:2013 standard on days 4 to 6 by the detection of total bacteria count on PCA (Plate Count Agar) from the milk samples of selected quarters (N = 10 on each day) from all cows. For the total bacteria count (TBC) on PCA, samples, and a series of 10-fold dilutions plated on PCA were set to obtain 15 TBC results (logarithmic (log) CFU/mL). Before plating, samples were 10-times diluted in 0.9% saline solution. Plating was performed using automatic plater Easy Spiral (Interscience, Saint-Nom-la-Bretèche, France).

### 4.5. Immunocytochemistry

In total, 135 samples of milk sediments were prepared at RIC Pro-Akademia, Poland, for further investigation. These were constituent of 3 repetitions (day 4, day 5, and day 6) with the collection of 100 mL of milk from the 15 cows of a selected quarter of each. Milk samples of 100 mL were obtained directly after the milking. In total, 300 mL of milk was prepared for centrifugation. Milk sediments were prepared by centrifugation of milk samples at 2000 (1000 to 3000) RPM for 3 min. The supernatant was carefully removed and discarded. Samples were again centrifuged at 2000 (1000 to 3000) RPM for 3 min. The supernatant was carefully removed and discarded. Samples were again centrifuged at 2000 (1000 to 3000) RPM for 3 min. The samples were centrifuged using Frontier 5000 Multi Pro (Model FC5816R, OHAUS, Parsippany, NJ, USA) laboratory centrifuge, equipped with a swing rotor (Model OH_30314828, OHAUS, Parsippany, NJ, USA). In total, 45 Eppendorf tubes of 5 mL were prepared with 2 mL of milk sample (sediment) and 2 mL of Tyrode’s solution buffer from each day of sample collection round. Samples were stored at −20 °C. The delivery of samples was carried out with dried ice.

Preparation and staining of milk smears were performed at the Laboratory of Morphology, Institute of Anatomy and Anthropology, Riga Stradins University. A modified biotin-streptavidin immunocytochemical method was performed to examine cytokines IL-1α, IL-4, IL-6, IL-12, IL-13, IL-17A, TNF-α, and IFN-γ in prepared milk smears [99,100].

For the cytological smear, fixation with the Stefanini (Zamboni) solution was performed [101]. Stefanini tissue fixative was prepared using paraformaldehyde (20 g), 150 mL picric acid (0.2% concentration), 425 mL Sorensen buffer (pH 7.2), and 425 mL distilled water. Paraformaldehyde was dissolved in distilled water. The resulting mixture was then placed in buffer and was added with filtered picric acid. Milk sediments were proceeded for a wash in TRIS buffer twice for 5 min. Samples were then placed in 3% hydrogen peroxide (H_2_O_2_) solution for 30 min at 4 °C. Washing with TRIS buffer was performed repeatedly for 5 min each time. Milk sediment samples were incubated with a HiDef DetectionTM reaction amplifier (code 954D-31, Cell MarqueTM, Rocklin, CA, USA) for the detection of antibodies acquired from mouse or rabbit. Incubation with HiDef DetectionTM horseradish peroxidase (HRP) polymer detector (code-954D-32, Cell MarqueTM, Rocklin, CA, USA) was managed. These procedures were substantially followed by washing the samples in a wash buffer and then processed with 3,3’-diaminobenzidine (DAB) Substrate Kit (code-957D-30, Cell MarqueTM, Rocklin, CA, USA) to obtain immunoreactive structure staining in brown color. To avoid the artifact presence, cells of smears were also stained by hematoxylin (code 05-M06002, Mayer’s, Bio Optica Milano S.p.A., Milano, Italy). The following antibodies were used for the immunocytochemical staining of milk smears: IL-1α (code ab7632, dilution 1:100, rabbit, Abcam, Cambridge, UK), IL-4 (code orb10908, dilution 1:100, rabbit, Biorbyt Ltd., Cambridge, UK), IL-6 (code sc-28343, dilution 1:50, mouse, Santa Cruz Biotechnology Inc., Santa Cruz, CA, USA), IL-12 (code orb10894, dilution 1:100, rabbit, Biorbyt Ltd., Cambridge, UK), IL-13 (code orb10895, dilution 1:200, rabbit, Biorbyt Ltd., Cambridge, UK), IL-17A (code ab79056, dilution 1:200, rabbit, Abcam, Cambridge, UK), TNF-α (code ab6671, dilution 1:100, rabbit, Abcam, Cambridge, UK), and IFN-γ (code ab218426, dilution 1:100, mouse, Abcam, Cambridge, UK).

Milk sediment samples were examined by using bright-field microscopy with a Leica DC 300F camera microscope (Leica DM500RB, Leica Biosystems Richmond, Buffalo Grove, IL, USA) for cytological analysis and photography. Acquired images were analyzed using Image-Pro Plus 6.0 software (Media Cybernetics, Rockville, MD, USA).

### 4.6. Data Statistical Analysis

In the data statistical analysis, SCC absolute values were transformed to logarithmic (log) 10 values (log10).

Immunoreactive cells were assessed by the semiquantitative counting method using the following labeling system set on the intensity: “−“—no immunoreactivity (Figure 3C; red arrow), “+”—weak immunoreactivity (Figure 3B; bottom red arrow), and “++”—intensive immunoreactivity (Figure 3B; top red arrow) [100]. Direct cell counts of immunoreactive and negative cells were estimated in absolute numbers [102]. The presented data are mean values (%) (immunocytochemistry) or mean values (SCC; TBC) (standard deviation (SD), multiplier) unless stated otherwise.

For the data analysis, the selected 15 cows were grouped by the status of healthy, subclinical mastitis-affected, and clinical mastitis-affected, with 5 subjects in each. ANOVA statistical tests with post-hoc tests were performed to test the significance of the time point and status of cows on immunoreactive cell counts. The dependent variable was immunoreactive cell count, whereas independent variables were the time period of day 4 to 6 (D) and the status of cows (S) with the division of study subjects into healthy cows, subclinical mastitis-affected cows, and clinical mastitis-affected cows. One-way repeated-measures ANOVA was performed to determine the interactions of time points to immunoreactive cell count in each status separately. Mixed two-way ANOVA was performed to determine the interactions of both independent variables on the dependent variable where subjects of each status are different. Here, within-subjects factors are time points, whereas subjects are separated into different groups based on between-subjects factor of status. Before the ANOVA tests, statistical test assumptions were tested. Outliers as assessed by examination of studentized residuals for values greater than ±3 were dealt with regarding the results and conclusions of statistical tests performed with/without the outlier. If the results and conclusions did not differ sufficiently, outliers were kept in the data, whereas outliers affecting the assumptions were dropped. Furthermore, Shapiro–Wilk test of normality, Levene’s test for homogeneity of variances, as well as Mauchly’s test of sphericity were performed accordingly. In a one-way repeated-measures ANOVA, multiple paired-samples t-tests with a Bonferroni adjustment for multiple comparisons were run for post-hoc tests. In mixed two-way ANOVA, post-hoc multiple comparisons for observed means were performed with a Tukey test for equal variances assumed, but Games–Howell test for equal variances was not assumed. Statistical simple main effects were examined if the interaction was statistically significant, but the main effects if the interaction was not statistically significant.

The statistical analysis was performed using the statistical program SPSS Statistics, version 26.0 (IBM Company, Armonk, NY, USA). In all the statistical analyses, two-tailed *p*-values < 0.05 were considered statistically significant.

## 5. Conclusions

In healthy cows, high numbers of immunoreactive cells substantiate host immunity under continuous pathogen exposure. Consequently, both local cellular and humoral immune responses play a role in defense responses, where pronounced findings of IL-1α, IL-4, IL-6, IL-12, IL-17A, and INF-γ highlight these suggestions in terms of relative health.

Together, data about naturally occurring subclinical and clinical mastitis demonstrate inflammatory responses to intramammary infection driven by IL-1α, IL-4, IL-12, IL-17A, and IFN-γ in subclinical mastitis, and IL-1α, IL-4, IL-6, and IL-17A in clinical mastitis. Moreover, host defense response in mastitis is characterized by the continuation or resolution of initial inflammation as no prominent increase or peak was evaluated in the immunoreactivity of proinflammatory factors. The latter approach may be substantiated by the protection of the udder, reducing transiently the risk of infection and lowering the severity of eventually occurring mastitis as microbiological examination noted a decrease in total bacterial count.

The understanding of the immune mechanisms involved in the mammary gland defense against invading microbiological pathogens may lead to the development of improved diagnostic approach and treatment, in particular, using cytokines to build diagnostic tools and design immunomodulatory strategies for the control of bovine mastitis. In this matter, IL-12 and INF-γ immunoreactivity was recognized to differ mastitis cases from the relative health status.

## Figures and Tables

**Figure 1 pathogens-11-00372-f001:**
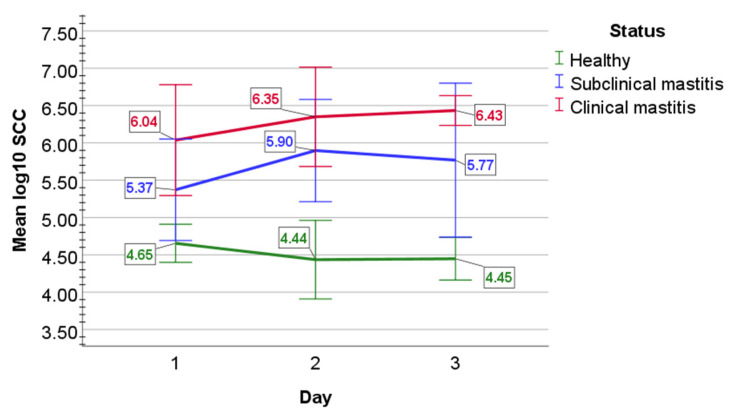
The graphical appearance of mean log10 SCC in the milk of healthy cows (green line) and cows with subclinical (blue line) and clinical mastitis (red line). Error bars: +/−2 SD.

**Figure 2 pathogens-11-00372-f002:**
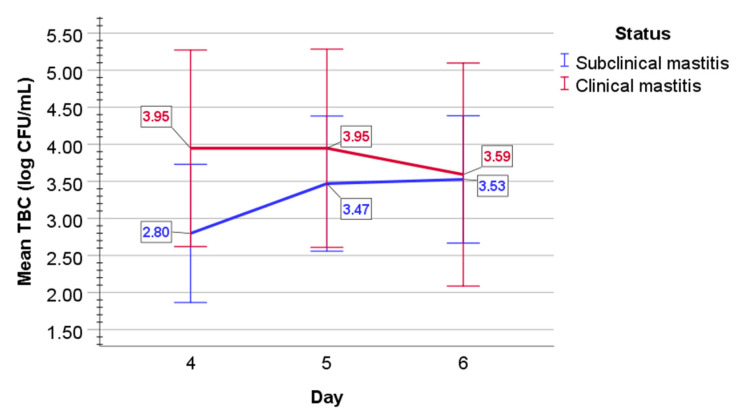
The graphical appearance of mean TBC (log CFU/mL) in the milk of cows with subclinical (blue line) and clinical mastitis (red line). Error bars: +/−2 SD.

**Figure 3 pathogens-11-00372-f003:**
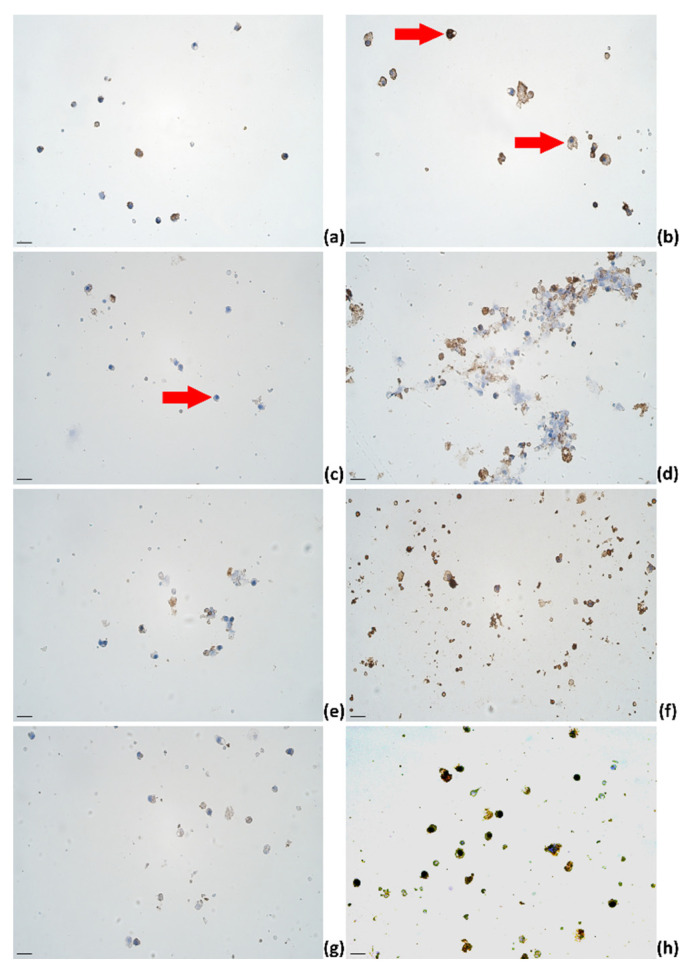
The appearance of (**a**) IL-1α, (**b**) IL-4, (**c**) IL-6, (**d**) IL-12, (**e**) IL-13, (**f**) IL-17A, (**g**) TNF-α, and (**h**) IFN-γ immunoreactive cells (epithelial cells (**a**–**d,f**,**g**); lymphocytes (**a**,**b**,**d**,**e**–**h**); macrophages (**a**,**b**,**d**,**e**–**h**)) in milk smears of healthy animals; IHC × 400. Scale bar: 20 μm.

**Figure 4 pathogens-11-00372-f004:**
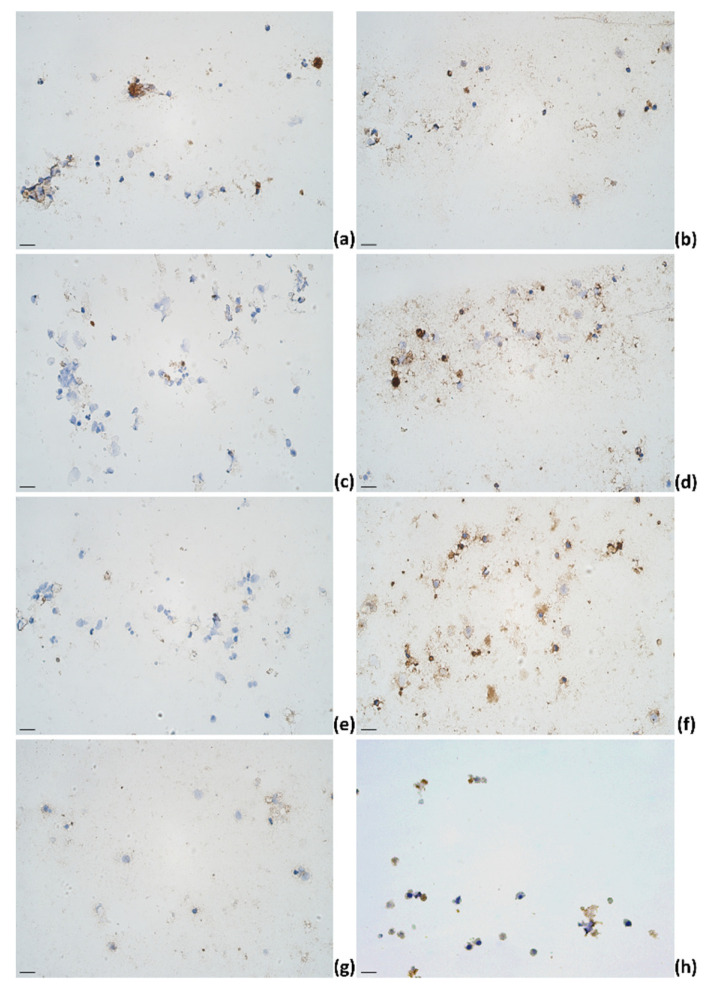
The appearance of (**a**) IL-1α, (**b**) IL-4, (**c**) IL-6, (**d**) IL-12, (**e**) IL-13, (**f**) IL-17A, (**g**) TNF-α, and (**h**) IFN-γ immunoreactive cells (epithelial cells (**f**,**g**); lymphocytes (**a**,**b**,**d**,**f**,**h**); macrophages (**a**,**b**,**d**,**f**); neutrophils (**b**,**d**,**f**)) in milk smears of subclinical mastitis-affected animals; IHC × 400. Scale bar: 20 μm.

**Figure 5 pathogens-11-00372-f005:**
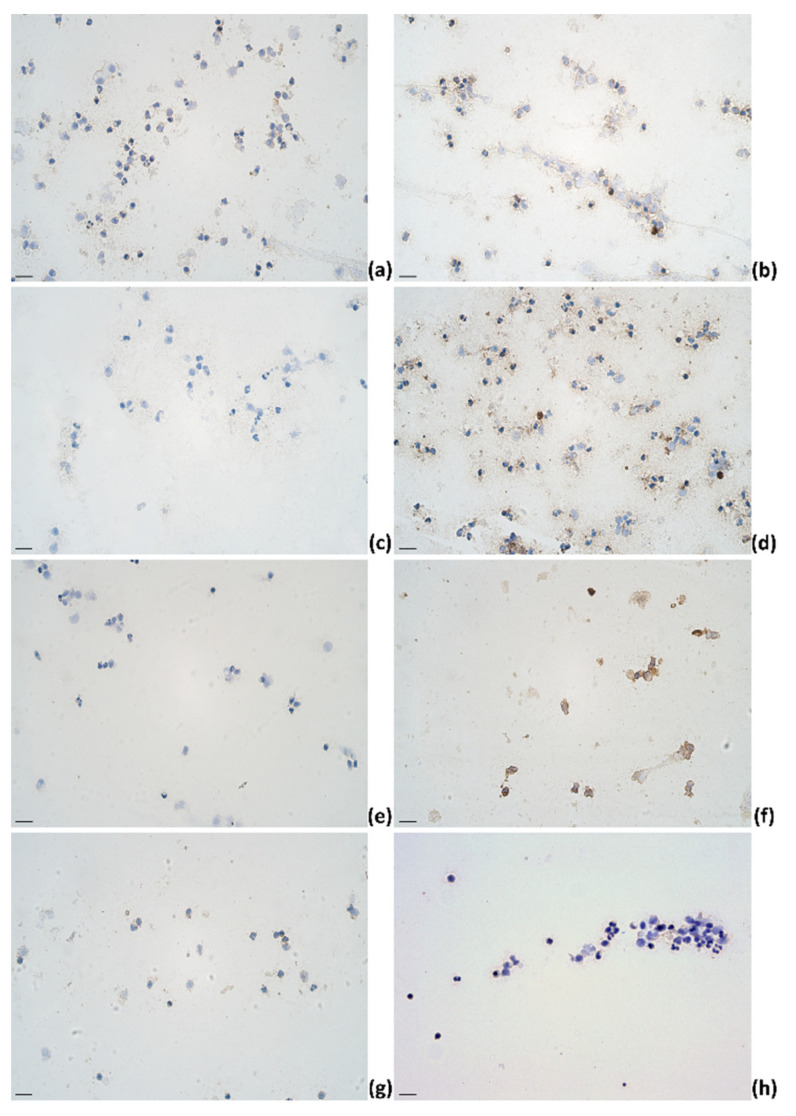
The appearance of (**a**) IL-1α, (**b**) IL-4, (**c**) IL-6, (**d**) IL-12, (**e**) IL-13, (**f**) IL-17A, (**g**) TNF-α, and (**h**) IFN-γ immunoreactive cells (epithelial cells (**b**,**f**); lymphocytes (**a**,**b**,**d**,**h**); macrophages (**a**,**b**,**d**,**f**,**g**); neutrophils (**a**,**b**,**d**)) in milk smears of clinical mastitis-affected cows; IHC × 400. Scale bar: 20 μm.

**Figure 6 pathogens-11-00372-f006:**
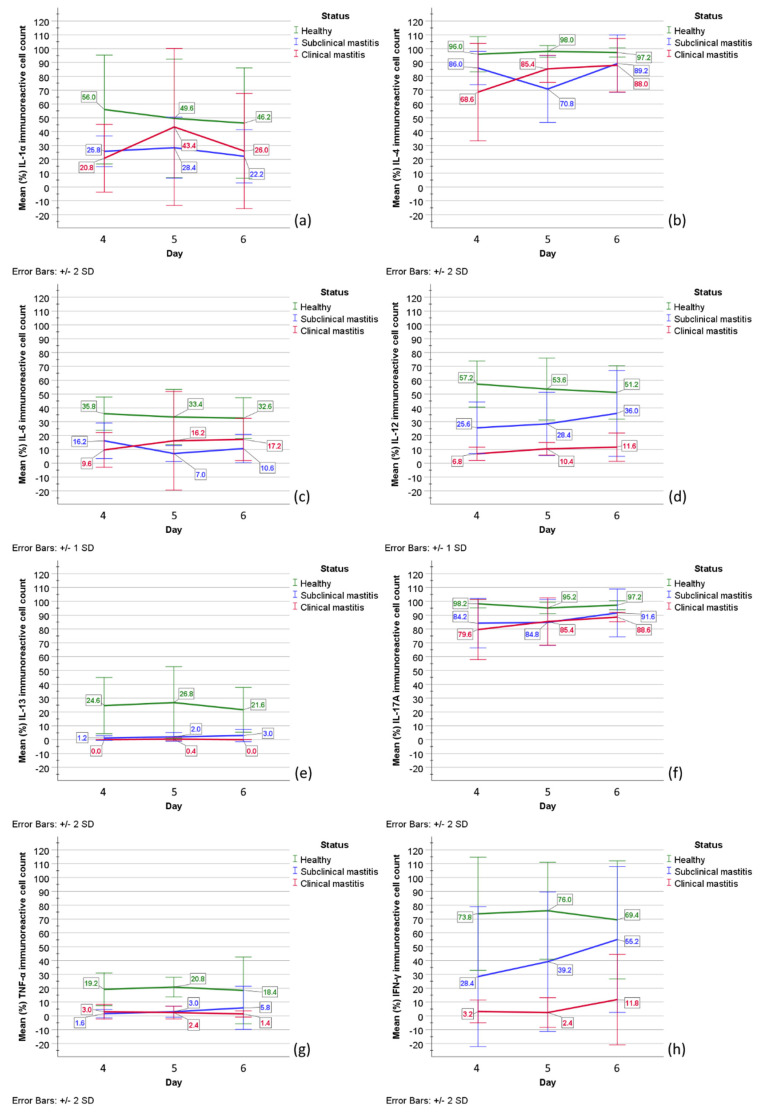
The graphical appearance of mean immunoreactive cell numbers (%) in healthy cows (green line) and cows with subclinical (blue line) and clinical mastitis (red line): (**a**) IL-1α; (**b**) IL-4; (**c**) IL-6; (**d**) IL-12; (**e**) IL-13; (**f**) IL-17A; (**g**) TNF-α; (**h**) IFN-γ. Together mean values (%) (SD) of intensively (++) and weakly (+) stained immunoreactive cells are demonstrated. Abbreviations in figure: IL-1α, IL-4, IL-6, IL-12, IL-13, IL-17A—interleukins (IL)-1α, -4, -6, -12, -13, -17A; TNF-α—tumor necrosis factor-alpha; IFN-γ—interferon-gamma. Error bars: +/- 1 SD; +/- 2 SD.

**Table 1 pathogens-11-00372-t001:** The appearance and mean numbers of IL-1α, IL-4, IL-6, IL-12, IL-13, IL-17A, TNF-α, and IFN-γ immunoreactive cells in healthy cows and cows with subclinical and clinical mastitis.

	**IL-1α**									**IL-4**									**IL-6**									**IL-12**								
	Day 4			Day 5			Day 6			Day 4			Day 5			Day 6			Day 4			Day 5			Day 6			Day 4			Day 5			Day 6		
	++	+	0	++	+	0	++	+	0	++	+	0	++	+	0	++	+	0	++	+	0	++	+	0	++	+	0	++	+	0	++	+	0	++	+	0
	Healthy																																			
Mean (%)	15	41	44	15	34	51	14	33	54	51	45	4	64	34	2	54	43	3	18	18	64	18	16	67	16	17	67	29	28	43	28	26	46	21	30	49
+/−	56 */44			49/51			47/54			96/4			98/2			97/3			36/64			34/67			33/67			57/43			54/46			51/49		
	Subclinical mastitis																																			
Mean (%)	6	20	74	8	21	71	5	17	78	24	64	12	15	56	29	31	55	14	5	11	84	3	4	93	1	9	89	10	16	74	12	16	72	14	22	64
+/−	26/74			29/71			22/78			88/12			71 */29			86/14			16/84			7/93			10/89			26/74			28/72			36/64		
	Clinical mastitis																																			
Mean (%)	1	20	79	6	37	57	3	23	74	19	56	31	17	69	15	23	65	12	1	9	90	6	10	84	2	15	83	2	5	93	3	7	90	2	10	88
+/−	21/79			43/57			26/74			75/31			86/15			88/12			10/90			16/84			17/83			7/93			10/90			12/88		
	**IL-13**									**IL-17A**									**TNF-α**									**IFN-γ**								
	Day 4			Day 5			Day 6			Day 4			Day 5			Day 6			Day 4			Day 5			Day 6			Day 4			Day 5			Day 6		
	++	+	0	++	+	0	++	+	0	++	+	0	++	+	0	++	+	0	++	+	0	++	+	0	++	+	0	++	+	0	++	+	0	++	+	0
	Healthy																																			
Mean (%)	4	20	76	4	23	73	3	19	79	17	81	2	21	74	5	23	74	3	1	18	81	2	19	79	2	16	82	42	31	26	42	43	24	35	35	30
+/−	24/76			27/73			22/79			98 * /2			95/5			97/3			19/81			21/79			18/82			73/26			76/24			70/30		
	Subclinical mastitis																																			
Mean (%)	0	1	99	0	2	98	0	3	97	10	74	16	11	74	15	17	75	8	0	2	98	0	3	97	0	6	94	12	16	72	16	23	60	14	42	49
+/−	1/99			2/98			3/97			84/16			85/15			92/8			2/98			3/97			6/94			28/72			40/60			26/49		
	Clinical mastitis																																			
Mean (%)	0	0	100	0	0	100	0	0	100	3	77	19	5	81	14	7	82	11	0	3	97	0	2	98	0	1	99	1	4	95	1	2	98	4	8	88
+/−	0/100			0/100			0/100			80/19			86/14			89/11			3/97			2/98			1/99			5/95			3/98			12/88		

Mean—mean value (%) of intensively (++), weakly (+) and negatively (0) stained immunoreactive cells, numbers are indicated with zero decimals; +/−—immunoreactive cells (both values of intensively (++) and weakly (+) stained immunoreactive cells) (+) vs. negative cells (no immunoreactivity) (−); numbers are indicated with zero decimals. Abbreviations in table: *—statistically significant difference (*p* < 0.05); IL-1α, IL-4, IL-6, IL-12, IL-13, IL-17A—interleukins (IL)-1α, -4, -6, -12, -13, -17A; TNF-α—tumor necrosis factor-alpha; IFN-γ—interferon-gamma.

**Table 2 pathogens-11-00372-t002:** Statistically significant interactions of time and status on mean numbers (%) of IL-1α, IL-4, IL-6, IL-12, IL-13, IL-17A, TNF-α, and IFN-γ immunoreactive cells in the healthy cows and animals with subclinical and clinical mastitis: a summary.

Cytokine	*p*-Value				
	D			S	S × D
	Healthy	Subclinical Mastitis	Clinical Mastitis		
IL-1α	0.253	0.294	0.068	0.071	0.019 *
IL-4	0.728	0.085	0.125	<0.001 *	0.031 *
IL-6	0.892	0.081	0.65	0.073	0.896
IL-12	0.614	0.494	0.557	0.004 *	0.852
IL-13	0.442	0.147	0.111	<0.001 *	0.909
IL-17A	0.007 *	0.084	0.374	0.015 *	0.063
TNF-α	0.842	0.294	0.255	<0.001 *	0.759
IFN-γ	0.586	0.156	0.178	<0.001 *	0.184

D—day (4, 5 and 6); S—status (healthy cows, subclinical mastitis-affected cows, clinical mastitis-affected cows); IL-1α, IL-4, IL-6, IL-12, IL-13, IL-17A—interleukins (IL)-1α, -4, -6, -12, -13, -17A; TNF-α—tumor necrosis factor-alpha; IFN-γ—interferon-gamma; *—statistically significant difference (p < 0.05).

## Data Availability

The data sets used and/or analyzed during the current study are available from the corresponding author on request.

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
