# Peer review of "Identification of Inflammatory and Regulatory Cytokines IL-1α-, IL-4-, IL-6-, IL-12-, IL-13-, IL-17A-, TNF-α-, and IFN-γ-Producing Cells in the Milk of Dairy Cows with Subclinical and Clinical Mastitis"

_pathogens, 2022, doi:10.3390/pathogens11030372_

Round 1

Reviewer 1 Report

Mastitis in cows is caused by the simultaneous occurrence and overlap of many unfavorable factors, of which the most important are pathogens. The etiological agents that cause mastitis may vary depending on the climate, species, breeding. These agents include a wide range of Gram-positive and Gram-negative bacteria, mycoplasmas, fungi, yeasts, algae and viruses. The role of cytokines in the pathogenesis of mastitis in cows is very important.

In my opinion It would be better if researchers tested milk with mastitis caused by one etiological agent (e.g. S. aureus or another pathogen). After all, the authors write in line 507:   „suggesting the late  onset of these factors depend on causative pathogens”.  

Line 545- 627    no information on the number of quarters with mastitis (one or more quarters for one cow?) Probably one quarter was affected  in each cow with mastitis…..

Line 615 - 627 

Chromagar mastitis is a medium that can be used by veterinarians for the rapid and simple differentiation of the key microorganisms involved in mastitis infections  but cannot be used for scientific research.   Chromagar mastitis media do not detect Streptococcus agalactiae, Enterococcus sp., coryneform bacteria and most of all  this plate  do not detect coagulase-negative staphylococci (CNS), which are one of the etiological agents of mastitis in cows. From yeast-like fungi, they detect only Candida albicans, which almost never  causes mastitis mycotica (mastitis mycotica is  caused by NCA - not Candida albicans). Chromagar mastitis  are completely useless in mastitis prototheca.

Reviewer 2 Report

This paper discusses host defense response cytokines in naturally occurring clinical and subclinical mastitis in dairy cattle. It is well written with a clear scope.

615: It is not clear which milk samples where bacteriologically tested; Did you tested healthy cows as well for carrier state?

Table 1: the title of the table is confusing

Round 2

Reviewer 1 Report

Introduction

I appreciate that the authors mention about the diversity of the etiological agents of bovine mastitis (also the etiology of mastitis in cows in Poland ) However, I strongly recommend to provide some references documenting the significance of each of these gropus (bacteria - algae - fungi ). As for the algae, I have a particular expertise in, I ask the authors to cite some of the most recent, large-scale studies (e.g.  https://pubmed.ncbi.nlm.nih.gov/30891936/)

I would rather not overestimate the diagnostic relevance of Chromagar mastitis agars. These media have many limitations. I believe they outnumber their advantages. So, what I propose is to disregard this aspect (microbial culture of mastitis pathogens) of the study, since themethodology is far from appropriate.

So I suggest you delete lines, text and table (everything is yellow marked   in the attached manuscript)

Lines number 548-577 (discussion), 157-165 (results), 1026-1041, 1052-1054 (references)

table no.2 (results)

text (as below)

Bacteriological examination

 Microbiological analysis was performed according to ISO 4833:2013 standard on day  4 to 6 by the detection of total bacteria count on PCA (Plate Count Agar), Chromagar Mastitis Gram positive (G+), and Chromagar Mastitis Gram negative (G-) from the milk  samples of selected quarters (N = 10 on each day) from all cows. For the total bacteria  count (TBC) on PCA, samples and series of 10-fold dilutions plated on PCA were set to  obtain 15 TBC results (logarithmic (log) CFU/mL). Furthermore, samples and series of  their 4 10-fold dilutions plated on Chromagar Mastitis G+ and G- were set to investigate  occurrence of several Gram positive and negative bacteria species (logarithmic CFU/mL).  Before plating, samples were 10-times diluted in 0.9 % saline solution. Plating was performed using automatic plater Easy Spiral (Interscience, France). The following species  were determined in milk samples of mastitis-affected cows: Streptococcus agalactiae, Streptococcus uberis, Staphylococcus aureus, Escherichia coli, and altogether Klebsiella, Enterobacter,  Citrobacter spp.

 CHROMagar™ Mastitis GP allows for isolation and differentiation of S. agalactiae, S. uberis, S. aureus. CHROMagar™ Mastitis GN allows for isolation and differentiation of E.  coli, Klebsiella, Enterobacter, Citrobacter, Proteus, Pseudomonas, and C. albicans. Presence of different microorganisms in the culture is observed through the observed colors, e.g. blue-  green for S. agalactiae, red for E. coli, white for C. albicans [98].

This correction will not affect the meaning of the work in any way.

I suggest the authors to discuss in their work a very recent paper by Bochniarz et al. (https://pubmed.ncbi.nlm.nih.gov/34944383/) on some biochemical aspects of mastitis milk.

Only to Authors from Poland.

In the future, before starting research on milk pathogens, it may be ask the advice of the following people  who do research mastitis (in alphabetical order)

Dr. Zofia Bakuła, University of Warsaw

Dr. Mariola Bochniarz,  University of Life Sciences in Lublin

MSc. Mateusz Iskra, University of Warsaw

Dr. Tomasz Jagielski,  University of Warsaw

Dr. Henryk Krukowski,  University of Life Sciences in Lublin

Dr. Henryka  Lassa, the Milk Research Laboratory in Bydgoszcz

Round 3

Reviewer 1 Report

Please add  a few publications about  bacterial and fungal agents of mastitis (also about the etiological agents  of mastitis in Poland,  because your research was performed in Poland http://www.medycynawet.edu.pl/archives/423/6339-summary-med-weter-76-01-6339-2020) to this sentence (line 54-55) "Mastitis is mainly a bacterial infection, however, in some cases is caused by algae, viruses, fungi or improper milking procedures [numbers from to]".  

Author Response

This manuscript is a resubmission of an earlier submission. The following is a list of the peer review reports and author responses from that submission.

Round 1

Reviewer 1 Report

The reviewer appreciates the opportunity to critique the submitted manuscript. Please, see attached document for specific comments. Overall, there needs to be more description of animals, how they were included or exclude in the study, and justification of sample size. The is a problem with classification of animals, particularly into the "sub-clinical mastitis" group. The fact that some of the animals in this group are described as showing mild swelling and other signs of inflammation means that these animals were actually clinical. Perhaps, it is better to have classification of mastitis based on clinical severity scores.

Reviewer 2 Report

The manuscript entitled “Identification of inflammatory and regulatory cytokines IL-1α, IL-4, IL-6, IL-12, IL-13, IL-17A, TNF-α, and IFN-γ in milk of dairy cows with subclinical and clinical mastitis” summarized the inflammatory and regulatory cytokines in milk of dairy cows with subclinical and clinical mastitis.  Generally, the manuscript is interesting. This paper has several weaknesses and needs improvement before publication.

This manuscript has major language problems. There are too many for me to modify them all. Authors are strongly encouraged to seek a native English speaker who may assist you modifying the document.

Comments:

  1. Insert the scale in all figures.
  2. Describe the items (treatment names) in abbreviation and below the all tables.
  3. Insert the P-value for the traits in all tables.
  4. Summarize the abstract, focus on the main findings and mention the small conclusion in at the end of abstract
  5. In the Introduction focus on the objectives and insert a few new reference and relevant findings

Cite following paper in introduction part

Wang, X.P., Luoreng, Z.M., Zan, L.S., Raza, S.H.A., Li, F., Li, N. and Liu, S., 2016. Expression patterns of miR-146a and miR-146b in mastitis infected dairy cattle. Molecular and cellular probes30(5), pp.342-344.

Song, N., Wang, X., Gui, L., Raza, S.H.A., Luoreng, Z. and Zan, L., 2017. MicroRNA-214 regulates immunity-related genes in bovine mammary epithelial cells by targeting NFATc3 and TRAF3. Molecular and cellular probes35, pp.27-33.

  1. Material and method needs to clarifying and summarizing- some detailed needs
  2. The subtitles in the material and method needs to summarizing Ethical approval and references must be mentioned in M&M
  3. which stage of somatic cells the immunocytochemical was performed?
  4. why not perform western blot analysis using the same antibody to analyze the protein level during different stages of adipocytes which will further explore role
  5. Was DNA extracted from whole blood collected at harvest? Why not collect a more sterile sample prior to harvest?

In conclusion, the research presented is interesting, well planned and carried out. The manuscript can still be improve revise by a native English speaker. Nevertheless, I believe that this work deserves publication in Animals after the inclusion of corrections.

Reviewer 3 Report

In their manuscript, the authors present the detection of cytokines reactive cells from healthy and mastitic milk. Diagnostic is still a very important point for treatment/ management of bovine mastitis and are more tools are needed.

Was there a statistical analysis done to verify that SCC, bacterial counts were different between the 3 groups of cows? This is a prerequisite to the whole study. Please present these results in the text and in the figures 1 and 2. How do you explain that healthy quarters have equivalent bacterial count at day 3 (fig 2)? How quarters with pathogens can be categorized as healthy? How cytokines expression can be compared in such conditions? This is a major flaw of the manuscript.

Cell numbers and populations vary according to the health status of the quarter. How this was taking into consideration in your evaluation of reactive cells? Were the immunoreactive cell numbers normalized between healthy, subclinical and clinical samples ?

What were your criteria to classify weak or intense immunoreactivity for the cells ?

Please describe the pathogens (at the species level) that were identified in the different cases of mastitis as the immune response can differ according to the pathogen.

What were the thresholds used for SCC between subclinical and clinical mastitis. SCC must be presented as log10.

Because subclinical means absence of symptoms can quarters with “mild swelling or inflammation symptoms” be categorized as subclinical ?

Please review the numbering of the days to clarify the manuscript.. In paragraph 2.2, days 1 to 3 seems to be the animal selection days while 4 to 6 are the analysis samples. In paragraph 2.2 and the rest of the manuscript, I guess that days 1 to 3 are the days 4 to 6 also ?

How were your milk samples done ? Were they aseptic ? How do you explain that healthy quarters have in average 2.588 log10 cfu/ml (fig 2) ?

Please precise if any treatment were given to the cows with mastitis.

I would suggest to remove L272-351, and to present figures 3, 4 and 5 and table 1 as supplementary data.

Specific comments :

Title: Please consider changing it as your study is about detection of cytokines producing cells in milk.

L118: What do you mean with “and 38 cows before milking” ? Heifers ?

Paragraph 2.1: This is confusing… Please clarify to make this paragraph more in agreement with paragraph 2.2.

L148: “their 54 10 fold”. I guess a correction is needed with 54.

L163 and following: RPM. This unit depends on the centrifuge and the rotor, please provide the information or provide the number of g that is more universal.

L200: Do you log10 transformed data (SCC, CFU) before statistical analysis. This is a requirement.

Section 3.1: This is an improper way to present SCC, please use log10 values in the text, in the graph.

Section 3.3 Were statistical analysis performed between -, +, ++ immunoreactive cells ? If yes please present it clearly.

L452 : Is this statistically significant ? Please used log10 transformed data before giving the folds of increases. This would be more biologically relevant. At day one, SCC for subclinical cows is 5.46 log10 cells while it is 4.67 log10 for healthy cows, a fold increase of 1.16.

L463-464: If there are pathogens, this is an intramammary infection and then the quarter cannot be considered healthy… The healthy quarters cannot be used as a comparator to the other 2 groups.

Figures 1 and 2 : Please add the statistical analysis.

Figure 1: Please check the y axis. Use numerical number to present the log10 values. At the moment the value in the boxes don’t match with the number on the y scale : 46789 between 1 and 25 (2.5E1).+

Figure 6: please precise that this is + and ++ immunoreactive cells. Please add the SD to the graphs and remove the table 2 that presents exactly the same data.

Table 3 : the header S above the 3 groups (Healthy, subclinical, clinical) is confusing. Please replace factor by cytokines.

Round 2

Reviewer 1 Report

Please, see attached responses. This reviewer thanks the authors for their revisions. However, the reviewer also has concerns raised in the attached document.

Reviewer 3 Report

Thank you for providing this improved version of your manuscript.

Please provide more information about the milk samplings. Was the foremilk discarded before taking the sample ? This foremilk discard allows to wash out all the bacteria that can colonize (or be present) the teat canal. Improper milking samplings could explain your bacterial count observed in the health cows.

I'm also wondering if your analyses were done on quarter sample or composite samples (mix of the milk from all quarters). Is it because it is a mix sample that you identify multiple pathogens in the cow ? According to Table 1: 5 of the healthy cows have S. aureus, Strep uberis,

I'm sorry to insist on this point but how can you qualify cows harbouring S. aureus in their quarters of healthy (your 5 healthy cows, if I understood properly table 1)? If a pathogen is detected, this means that there is an infection. There is no sign so this qualifies for a perfect subclinical mastitis. These cows are infected so it can be hypothesized that there is a local immune response in the milk. So how then this group can be the control group ? I think that this study should have been designed according to the presence of pathogens (drivers of the immune response) rather than on the SCC (that is a marker, not always good of the infectious status of the mammary gland).